# SEMANTIC-ALIGNED QUERY SYNTHESIS FOR ACTIVE LEARNING

## ABSTRACT

Active learning (AL) reduces data annotation costs by querying labels from human annotators for the most informative unlabeled data points during model training. Existing AL methods generally assume the availability of a large amount of unlabeled samples for query selection. However, collecting raw data in practice can be expensive, even without considering the cost of labeling. Membership query synthesis circumvents the need for an unlabeled data pool by directly generating informative queries from the input space. Nevertheless, existing approaches often generate instances lacking semantic meaning, thereby increasing the difficulty of labeling. In this paper, we propose the Generative Membership Query Descriptor (GenMQD) method for AL to mitigate the risk of generating unrecognizable instances. The key idea is to generate textual descriptions of the desired data, instead of the data samples themselves. Then a pre-trained multi-modal alignment model (e.g., CLIP) can be leveraged to transform these features into natural language texts for data gathering purposes. Extensive experiments on image classification benchmark datasets against query synthesis state-of-the-art methods demonstrate that, on average, GenMQD can improve model accuracy by 2.43% when gathering and labeling 500 examples. A large-scale user study verifies that human oracles prefer GenMQD generated queries over generated image-based queries.

## 1 INTRODUCTION

Active learning (AL) (Settles, 2009) aims to reduce the cost of data annotation by selectively querying the most informative unlabeled data from an oracle, typically a human annotator. By focusing on the most valuable data points, AL methods can achieve the better model performance with fewer labeled samples (Hanneke, 2014). Most of current AL methods are designed for either the pool-based setting or the stream-based setting. For pool-based AL approaches, a large pool of unlabeled data is generally assumed to be available, from which a subset is selected for human annotation (Kirsch et al., 2019; Sener & Savarese, 2018; Tang & Huang, 2022). In contrast, the stream-based AL approaches receive data incrementally, requiring the AL algorithms to decide in real-time whether to query the oracle for the label of each received instance (Huang et al., 2024; Chen et al., 2022; Cacciarelli & Kulahci, 2024).

In practice, data collection can be prohibitively expensive in turns of effort and costs involved, and this can be further exacerbated by the cost of data annotation (Cohen et al., 2020; Jia et al., 2020; Aizawa et al., 2020). For example, acquiring high-quality medical imaging scans or gathering sensor data from remote locations in environmental monitoring, can be extremely resource-intensive. In addition, certain types of raw data, such as music concert recordings or artificially created manga images, require significant human effort to produce. In such scenarios, AL strategies requiring an existing pool of unlabelled samples can lead to considerable waste, as the majority of the collected data remains unlabeled, thereby resulting in inefficient utilization of resources and increased costs.

The sub-field of Membership Query Synthesis (Schumann & Rehbein, 2019; Guo et al., 2021; Tran et al., 2019), which belongs to AL methods that do not rely on pre-existing unlabeled pool of data, has emerged in recent years, but only receiving relatively less attention in the AL relevant literature. These methods either generate informative synthetic data (e.g., using Generative Adversarial Networks (GANs)), or produce feature vectors through optimization in the feature space, which are in turn used for querying the oracle. Although these approaches have demonstrated effectiveness in

simple scenarios (e.g., finite problem domains (Angluin, 2004)), they face a significant limitation: *the generated synthetic data often lacks semantic meaning and might not be easily recognizable by human annotators* (Lang & Baum, 1992). Furthermore, although existing AL methods are starting to adopt trained models as oracles to provide labels for the synthetic data, they might be less tractable in practice.

To address these challenges, we propose the Generative Membership Query Descriptor (`GenMQD`) method for AL. It generates textual descriptions of the desired data with rich semantic context, thereby providing transparent interaction between AL and the human oracle. Specifically, `GenMQD` first syntheses the desired data feature such that the model learned by the generated feature can maximize the model performance gain on the validation set. Mathematical analysis shows that this problem can be approximately solved by the influence function of perturbing of the training point (Koh & Liang, 2017). To further obtain descriptive text of the desired data, `GenMQD` is designed to leverage a pre-trained multi-modal alignment model (e.g., CLIP in Radford et al. (2021)) to describe the desired instance in natural language terms. With CLIP as an example, we demonstrate that the text embedding of the desired data is exactly its image embedding. Therefore, existing methods that map the CLIP image embedding or text embedding into the text descriptions can be adopted to generate the required information. Finally, the generated text descriptions can be used for querying the oracle to obtain a batch of instances that match the descriptions.

We have conducted extensive experiments on CIFAR-10 and iNaturalist 2021 benchmark datasets against membership query synthesis state-of-the-art approaches to evaluate the effectiveness of `GenMQD`. On average, it improves model performance by 2.43%, when labeling the same amount of data with the compared methods. A large-scale user study with 340 responses from human annotators verifies that human oracles prefer `GenMQD` generated queries over generated image-based queries. Compared to existing query synthesis methods, `GenMQD` offers a more human-friendly way to data gathering and mitigates the risk of generating instances which are difficult to annotate. The generated descriptions, even if do not correspond directly to real-world data, can still convey useful information to the oracle. These descriptions are designed to reveal the uncertainties in the knowledge within a target model, thereby guiding human oracles toward more actionable data gathering and annotation decisions.

## 2 RELATED WORKS

AL aims to reduce the cost of data labeling by selectively querying oracles for the labels of the most valuable data points. Common selection notions in AL include informativeness and representativeness (Settles, 2009; Ren et al., 2021). Informativeness refers to selecting data points for which the model is the most uncertain. Techniques for achieving this notion include estimating the distance between data and the decision boundary (Zhu & Bento, 2017); estimating the expected loss of unlabeled data (Yoo & Kweon, 2019); and estimating the prediction variance of the same instance with slight disturbing (Gal et al., 2017), etc. Representativeness, on the other hand, seeks to ensure that the selected data points reflect the broader distribution of the entire dataset, thereby mitigating bias and improving generalization. Techniques to achieve this goal include maximum mean discrepancy (Du et al., 2015), Coreset (Sener & Savarese, 2018), and variational autoencoder (VAE) (Sinha et al., 2019), etc.

Recently, there has been significant interest in combining multiple selection criteria, which usually lead to better performances. For example, Du et al. (2015) leverages informativeness and representativeness simultaneously to improve performance. Tang & Huang (2019) further incorporates ease of data evaluation to avoid querying overly-hard samples facing the learning model. Shui et al. (2020) proposes a principle to design deep AL algorithms, which combines both informativeness and representativeness.

Another AL paradigm, membership query synthesis, circumvents the need for selecting from a pool of unlabeled data. Instead, it queries oracles with arbitrary instances from the input space for labeling. This is usually achieved via data synthesis/generation (Zhu & Bento, 2017; Kong et al., 2019; Yan et al., 2020). Existing membership query synthesis methods can be broadly divided into two categories: 1) feature-based methods, and 2) generative methods. Feature-based methods (Angluin, 2004; Cohn et al., 1996; Yan et al., 2020) leverage predefined rules or criteria (e.g., uncertainty) to generate new data features from the input space based on the model's current state or predictions.

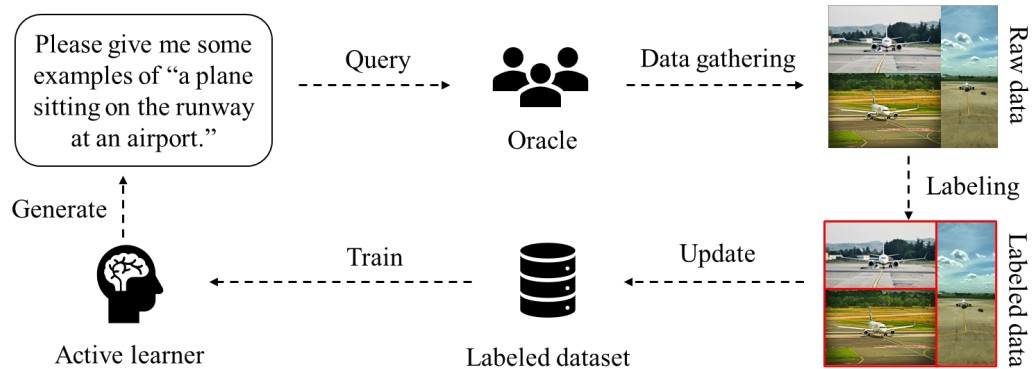

Figure 1: The overall framework of the proposed GenMQD method. In each iteration, AL generates a textual description of the desired data. Then, the oracle collects and labels a batch of data that matches the generated description. The acquired data is incorporated into the training set. The active learner is updated accordingly.

One significant drawback of these methods is the limited range of application, since they are hard to generalize to tasks in various domains (Angluin, 2004). Generative methods (e.g., VAE, GAN) can synthesize realistic data points that are likely to improve the model. A diverse range of methods have been proposed to generate desired data for annotation (Zhu & Bento, 2017; Tran et al., 2019). However, training an effective generation model requires plenty of data, which is usually unavailable in AL. Moreover, the generated instances either need to be annotated by a human oracle (Zhu & Bento, 2017; Schumann & Rehbein, 2019) or to be assigned pseudo-labels (e.g., by a class-conditional GAN (Kong et al., 2019)). They are prone to unrecognizable queries and uncertain noises. The proposed GenMQD method can address such limitation of existing generative membership query synthesis methods in AL area.

## 3 THE PROPOSED GenMQD METHOD

In this section, we introduce the proposed GenMQD method. The overall framework is illustrated in Fig. 1. The key idea is to generate textual descriptions of the desired data for data gathering and labeling. Next, we will introduce some preliminaries, followed by the detailed techniques.

### 3.1 PRELIMINARIES

Given a limited labeled dataset $L = \{z_i\}_{i=1}^{n_l}$ and a small validation set $V = \{z_i\}_{i=1}^{n_v}$, where $z_i = \{\boldsymbol{x}_i, y_i\}$, $\boldsymbol{x}_i$ denotes the $i$-th instance, which belongs to the input domain $\mathcal{X}$, $y_i \in \mathcal{Y}$ is its corresponding label, $n_l$ and $n_v$ are the sizes of $L$ and $V$, respectively. Our problem is to generate the text description $t(\boldsymbol{x}^*)$ for the desired instance $\boldsymbol{x}^*$. Then, an oracle will collect and label a small batch of data that matches the text description, denoted as $Q = \{z_i \mid z_i \sim p(t(\boldsymbol{x}^*)), i = 1, \ldots, n_q\}$ and $z_i \sim p(t(\boldsymbol{x}^*))$ represents sampling data from the latent distribution implicitly determined by $t(\boldsymbol{x}^*)$. Let $f(\cdot; \boldsymbol{\theta})$ denote the prediction model parameterized by $\boldsymbol{\theta}$. The learning objective is to optimize a loss function $\ell(\cdot)$ over the labeled data.

The influence function (Koh & Liang, 2017) evaluates the effect of a training data on the model prediction. Assuming that $\boldsymbol{x}$ is continuous, $\ell(\cdot)$ is differentiable in $\boldsymbol{\theta}$ and $\boldsymbol{x}$. As derived by Koh & Liang (2017), the influence of perturbation $\delta$ of a training point $z$, i.e., $z \mapsto z_\delta$ where $z_\delta \stackrel{\text{def}}{=} (\boldsymbol{x} + \delta, y)$, $\delta \to 0$, on the test point $z_{\text{test}}$, can be approximated as:

$$
\begin{aligned}
\mathcal{I}_{\text{pert,loss}}\left(z, z_{\text{test}}\right) &\stackrel{\text{def}}{=} \nabla_\delta \ell\left(z_{\text{test}}, \hat{\boldsymbol{\theta}}_{z_\delta, -z}\right)\Big|_{\delta=0} \\
&= -\nabla_{\boldsymbol{\theta}} \ell\left(z_{\text{test}}, \hat{\boldsymbol{\theta}}\right)^\top H_{\hat{\boldsymbol{\theta}}}^{-1} \nabla_{\boldsymbol{x}} \nabla_{\boldsymbol{\theta}} \ell\left(z, \hat{\boldsymbol{\theta}}\right),
\end{aligned}
\tag{1}
$$

where $\hat{\boldsymbol{\theta}}_{z_\delta, -z}$ is the empirical risk minimizer on the training points with $z_\delta$ in place of $z$ and $H_{\hat{\boldsymbol{\theta}}} \overset{\text{def}}{=}$ $\frac{1}{n_l} \sum_{i=1}^{n_l} \nabla_{\boldsymbol{\theta}^2} \ell\left(z_i, \hat{\boldsymbol{\theta}}\right)$ is the Hessian.

The CLIP model (Radford et al., 2021) aligns visual and textual representations through contrastive learning. For a given batch of image-text pairs, the goal of CLIP is to maximize the cosine similarity between the embeddings of corresponding pairs while minimizing the similarity between mismatched pairs. This is achieved by optimizing a contrastive loss. By training in this manner, CLIP effectively aligns visual and textual modalities, hence, enabling zero-shot image classification and text-based image retrieval tasks.

## 3.2 QUERY SYNTHESIS

`GenMQD` first syntheses the feature vector of a data point that is the most likely to improve the target model. To achieve this goal, we propose to optimize a randomly initialized feature vector such that the model trained with the generated feature vector can maximize the performance gain on the validation set. Formally, the objective function can be formulated as:

$$\arg\min_{z^*} \frac{1}{n_v} \sum_{z_i \in V} \ell(z_i, \hat{\boldsymbol{\theta}}_{L \cup z^*}), \tag{2}$$

where $\hat{\boldsymbol{\theta}}_{L \cup z^*}$ is the empirical risk minimizer on the dataset $L \cup z^*$ and $z^* = \{\boldsymbol{x}^*, y^*\}$. However, directly solving this optimization problem is NP-hard. To overcome this challenge, we further re-formulate our problem into the following form: given a desired class $y^*$, we add a randomly initialized instance $\boldsymbol{x}^*$ into the training set $L$ and iteratively optimize $\boldsymbol{x}^*$ by updating it with $\boldsymbol{x}^* + \delta$. In this way, the objective of one optimizing iteration can be formulated as:

$$\min_{\delta} \frac{1}{n_v} \sum_{z_i \in V} \ell(z_i, \hat{\boldsymbol{\theta}}_{z_\delta^*, -z^*}), \tag{3}$$

and the first order derivative of Eq. (3) is:

$$\frac{1}{n_v} \nabla_\delta \left[ \sum_{z_i \in V} \ell\left(z_i, \hat{\boldsymbol{\theta}}_{z_\delta^*, -z^*}\right) \right]. \tag{4}$$

Note that the derivative of Eq. (3) is exactly the form of the influence of perturbing of the instance $\boldsymbol{x}^*$ as defined in Eq. (1). Therefore, we leverage the influence function to approximately solve the optimization problem Eq. (3). The informative data feature can be generated by iteratively updating the randomly initialized instance $\boldsymbol{x}^*$ with a small step size as:

$$\frac{1}{n_v} \nabla_\delta \left[ \sum_{z_i \in V} \ell\left(z_i, \hat{\boldsymbol{\theta}}_{z_\delta^*, -z^*}\right) \right]$$

$$\approx \frac{1}{n_v} \sum_{z_i \in V} \nabla_\delta \ell\left(z_i, \hat{\boldsymbol{\theta}}_{z_\delta^*, -z^*}\right)\big|_{\delta=0} \tag{5}$$

$$= \frac{1}{n_v} \sum_{z_i \in V} \left[ -\nabla_{\boldsymbol{\theta}} \ell\left(z_i, \hat{\boldsymbol{\theta}}\right)^\top H_{\hat{\boldsymbol{\theta}}}^{-1} \nabla_{\boldsymbol{x}^*} \nabla_{\boldsymbol{\theta}} \ell\left(z^*, \hat{\boldsymbol{\theta}}\right) \right]. \quad \text{(by Eq. (1))}$$

In summary, the proposed query synthesis method begins by introducing a randomly initialized instance, denoted as $\boldsymbol{x}^*$, into the labeled set. Next, it iteratively updates $\boldsymbol{x}^*$ using the rule $\boldsymbol{x}^* \leftarrow \boldsymbol{x}^* + \alpha\delta$, where $\delta$ is an incremental adjustment optimized according to the gradient-based updating in Eq. (5). The optimization of $\delta$ is designed to maximize the model performance on the validation set after learning with the newly added instance. This iterative updating process continues until the feature vector $\boldsymbol{x}^*$ converges to a stable solution, at which point it is deemed adequately informative to be added as a synthesized query.

## 3.3 DESCRIPTION GENERATION

After generating the desired data point $z^*$, a pre-trained multi-modal alignment model is leveraged to describe $z^*$. Considering image classification tasks, CLIP (Radford et al., 2021) can be adopted

---

**Algorithm 1** The Proposed `GenMQD` Algorithm

---

**Input:** Labeled set $L$, validation set $V$, pre-trained CLIP model, classification model parameter $\boldsymbol{\theta}$, optimization epochs $T$, learning rate $\alpha$, query batch size $n_q$.

**Output:** Active collected dataset $Q$.

1: $Q \leftarrow \varnothing$
2: $\{\boldsymbol{x}_i \in L \cup V\} \leftarrow$ extract features for all images using a pre-trained CLIP image encoder.
3: **for** $y^* = \{1, \dots, |\mathcal{Y}|\}$ **do**
4:      $\boldsymbol{x}^* \leftarrow \text{Mean}(\{\boldsymbol{x}_i \in L \mid y_i = y^*\})$ ▷ *Initialize a feature vector using the class center of $y^*$.*
5:      **for** $j = 1, \dots, T$ **do**
6:          $\hat{\boldsymbol{\theta}} \leftarrow$ get the empirical risk minimizer on $L \cup \{\boldsymbol{x}^*, y^*\}$.
7:          $\delta \leftarrow \frac{1}{n_v} \sum_{z_i \in V} \left[ -\nabla_{\boldsymbol{\theta}} \ell \left( z_i, \hat{\boldsymbol{\theta}} \right)^\top H_{\hat{\boldsymbol{\theta}}}^{-1} \nabla_{\boldsymbol{x}^*} \nabla_{\boldsymbol{\theta}} \ell \left( z^*, \hat{\boldsymbol{\theta}} \right) \right]$      ▷ *By Eq. (5)*
8:          $\boldsymbol{x}^* \leftarrow \boldsymbol{x}^* + \alpha \delta$
9:      **end for**
10:     $t(\boldsymbol{x}^*) \leftarrow$ decode $\boldsymbol{x}^*$ into natural language as text description.
11:     $Q \leftarrow Q \cup \{z_i \mid z_i \sim p(t(\boldsymbol{x}^*)), i = 1, \dots, n_q\}$ ▷ *Collect data that matches the description.*
12: **end for**
13: **return** $Q$

---

to generate human-readable descriptions for active raw data gathering. Specifically, the desired data is first mapped into the latent image representation space of CLIP. This can be achieved using the pre-trained image encoder of the CLIP model (i.e., $\boldsymbol{v}_{\boldsymbol{x}^*} = \text{ImgEncode}(\boldsymbol{x}^*)$). Note that the CLIP image encoder and text encoder will normalize the embedding such that its $\ell_2$ norm equals to 1 (i.e., $\|\boldsymbol{v}_{\boldsymbol{x}^*}\|_2 = 1$).

Our problem has now become finding a text description $t(\boldsymbol{x}^*)$ such that its text embedding $\boldsymbol{t}_{\boldsymbol{x}^*} = \text{TxtEncode}(t(\boldsymbol{x}^*))$ best aligns the generated instance according to the pre-trained CLIP model. Recall that CLIP model maximizes the cosine similarity of paired image-text pairs during training. Therefore, the desired text embedding $\boldsymbol{t}_{\boldsymbol{x}^*}$ can be approximated by maximizing its cosine similarity with $\boldsymbol{v}_{\boldsymbol{x}^*}$ as:

$$\max_{\boldsymbol{t}_{\boldsymbol{x}^*}} \frac{\boldsymbol{t}_{\boldsymbol{x}^*}^\top \boldsymbol{v}_{\boldsymbol{x}^*}}{\|\boldsymbol{t}_{\boldsymbol{x}^*}\| \|\boldsymbol{v}_{\boldsymbol{x}^*}\|} \quad s.t. \quad \|\boldsymbol{t}_{\boldsymbol{x}^*}\| = \|\boldsymbol{v}_{\boldsymbol{x}^*}\| = 1 \,. \tag{6}$$

Since the $\ell_2$ norm of the text and image embeddings produced by CLIP equal to 1, the optimal solution of $\boldsymbol{t}_{\boldsymbol{x}^*}$ is $\boldsymbol{v}_{\boldsymbol{x}^*}$, which maximizes the inner product.

In this way, we have derived that the text embedding of the desired data is exactly its image embedding. To further transform the embedding into natural language, off-the-shelf methods can be leveraged to map the CLIP image embedding or text embedding into textual descriptions. Here, we adopt the method from (Li et al., 2023) to translate the CLIP embedding into texts. Finally, the oracle will collect and annotate a small batch of data that satisfies the text descriptions, and add them to the labeled set for model updating.

### 3.4 EFFICIENT IMPLEMENTATION

To improve the practicality of `GenMQD`, we introduce an efficient implementation in this section. To simplify computations, we leverage a pre-trained CLIP model with "ViT-B/32" backbone as the feature extractor. Specifically, the image encoder pre-processes the images. Subsequently, a 3-layer neural network $f(\cdot; \boldsymbol{\theta})$ is trained on the extracted features for classification. In this way, the input space aligns with the embedding space defined by the CLIP image encoder (i.e., $\boldsymbol{x} = \boldsymbol{v}_{\boldsymbol{x}}$), thereby eliminating the need to map the generated features into the CLIP embedding space. In addition, using a small neural network for prediction task facilitates the calculation of the influence function in query synthesis step. Furthermore, we initialize $\boldsymbol{x}^*$ using the class center of the target class $y^*$ to reduce the risk of the synthesized query being distant from the desired class and to expedite the query synthesis convergence. Finally, a query is generated for each class in each iteration for class balancing and batch-mode querying in our implementation. We summarize the main procedures of `GenMQD` for active data gathering by generating text description in Algorithm 1.

## 4 EXPERIMENTAL EVALUATION

### 4.1 EXPERIMENT SETTINGS

**Datasets.** We employ a commonly used image classification dataset - CIFAR-10 (Krizhevsky, 2009) - and a large dataset - iNaturalist 2021 (Grant Van Horn, 2021) - to study the effectiveness of the proposed `GenMQD` method. CIFAR-10 is a 10-class image classification dataset, consisting of 50,000 training images and 10,000 test images, with 6,000 images per class in total. It has been widely adopted in query synthesis studies (Kong et al., 2019; Zhu & Bento, 2017). iNaturalist 2021 contains about 2.7 million training images. Since the generative methods often struggle to generalize well to the datasets with a large number of classes, we use the super categories as the prediction target in this dataset. Specifically, there are 11 super categories in iNaturalist 2021. The validation set with 100,000 images is used as the test set in our experiments.

**Comparison Methods.** Since there are few methods for active text description generation, we compare our `GenMQD` method with state-of-the-art data synthesis methods and a text generation baseline. Specifically, the following approaches are included in our experiments:

1. RandomText: Querying fixed class-specific prompts for data gathering and labeling.

2. GAAL (Zhu & Bento, 2017): Generating images using GAN and queries the oracle about the ones near the decision boundary.

3. ACGAN (Odena et al., 2017): Using a class-conditional GAN to generate images with pseudo-labels.

4. ActiveGAN (Kong et al., 2019): Training a class-conditional GAN to generate uncertain images. The data generated by the trained GAN will be added to the labeled set.

5. `GenMQD`-Fea: A variant of the proposed method, which only uses the synthesized data point $z^*$ for model updating.

6. `GenMQD`: The proposed method. Generating textual descriptions of uncertain data for active data gathering and labeling.

Note that, `GenMQD` and RandomText generate texts for querying; GAAL, ActiveGAN and ACGAN train a class-conditional GAN over the labeled set for data synthesis to update the model; and `GenMQD`-Fea only uses the synthesized data point $z^*$ for model updating.

**Models.** We leverage a pre-trained CLIP model with "ViT-B/32" backbone as the feature extractor, followed by a 3-layer fully connected network as the classification model. To simulate the data gathering procedure by a human oracle, we employ a pre-trained Text-to-Image generative model, Stable Diffusion, to generate the data that matches the queried text description for model updating. Specifically, we use the implementation in Huggingface[1] to generate images. For the class-conditional GAN, we use the implementation in the repository[2] with the default parameters for all image synthesis methods.

**Training Strategy.** For our `GenMQD` method, the updating step size parameter $\alpha$ is set to $10^{-4}$, and the number of optimization epochs $T$ is set to 200. The open-sourced project DeCap (Li et al., 2023) is adopted to translate the CLIP embedding into texts for `GenMQD`. RandomText uses a template of "a photo of {class_name}" as prompts for image generation. GAAL requires a human oracle to annotate the generated data, which brings challenges to reproduce the results. For fair comparison, we replace the GAN with a class-conditional GAN in the same way as the other data synthesis methods to generate images with pseudo-labels. The source code is in the supplementary materials.

**AL Settings.** For each dataset, we uniformly sample $10\times$ number of classes from the training set as the initial labeled set. A validation set with the same size as the labeled set is sampled for `GenMQD`. In each iteration, the query size is set as the number of classes for all comparison methods. The model is updated on both the initial labeled set and the generated data. Then, it is evaluated on the test set for performance comparison. We report the learning curves reflecting changes in model accuracy with respect to query iterations.

---

[1]https://huggingface.co/stabilityai/stable-diffusion-2-base
[2]https://github.com/eriklindernoren/PyTorch-GAN

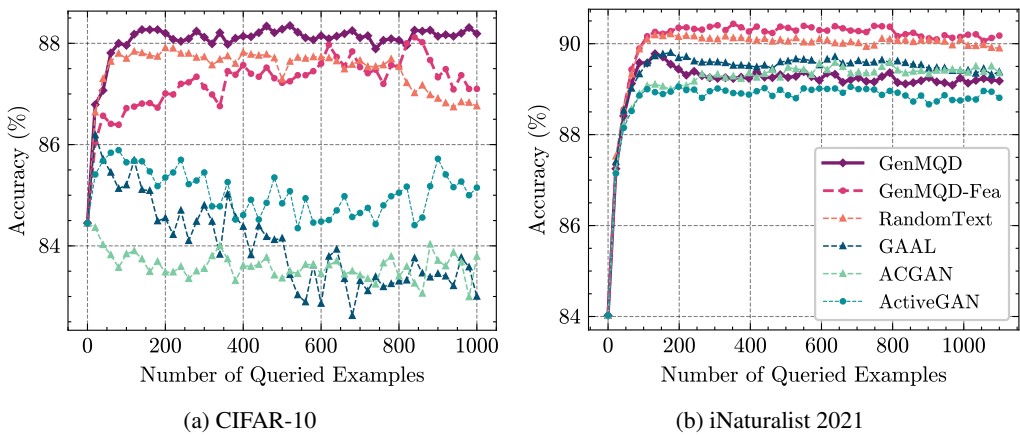

(a) CIFAR-10        (b) iNaturalist 2021

Figure 2: Performance comparison. (a) CIFAR-10; (b) iNaturalist 2021

## 4.2 RESULTS AND DISCUSSION

The learning curves of different methods compared on CIFAR-10 and iNaturalist 2021 datasets are presented in Fig. 2. The results on CIFAR-10 show that the proposed `GenMQD` method achieves the best accuracy. It consistently outperforms RandomText, indicating that the our generated texts are informative to the model. `GenMQD`-Fea achieves positive effect on model accuracy. However, the improvement is not as significant as `GenMQD`. This can be attributed to the fact that the pre-trained Stable Diffusion2 model introduces additional knowledge to facilitate model learning, thereby further improving model accuracy. GAAL, ActiveGAN and ACGAN are less effective in improving model accuracy. This can be atributed to the fact that the size of the initial labeled dataset is small, which might be insufficient to train an effective GAN model. Note that there is a common assumption in AL setting that the initial labeled set is small. ACGAN achieves the worst performance on CIFAR-10, as it does not filter any generated image, which might introduce noise. `GenMQD`-Fea surpasses all the image generation methods, suggesting that our proposed query synthesis method is capable of generating more informative data.

For iNaturalist 2021 dataset, our `GenMQD`-Fea method achieves the highest model accuracy performance. However, `GenMQD` does not perform well. This can be attributed to the incompatibility of the pre-trained text decoding model with the dataset, as the superiority of `GenMQD`-Fea suggests that the synthesized data point $z^*$ is informative and the performance of RandomText implies that the Text-to-Image model performs well. In our experiment, we employ an off-the-shelf method, DeCap (Li et al., 2023), to decode the CLIP embedding into text. More effective decoders can be selected according to the dataset to further improve the performance. We leave this matter for future study. GAAL, ActiveGAN and ACGAN effectively improve model accuracy with the generated images. However, their improvement is not significant compared to our proposed data synthesis method, demonstrating the superiority of the proposed `GenMQD` method.

## 4.3 STUDY ON DIFFERENT SIZES OF THE INITIAL LABELED SET

We conduct experiments using different sizes of the initial labeled set to evaluate the performance of our `GenMQD` method. Specifically, larger labeled sets with sizes of 500 and 1,000 are utilized across all comparison methods. The performance of the comparison methods on CIFAR-10 is reported in the upper part of Table 1.

It can be observed that as the size of the initial labeled dataset increases, the performance of different methods improves accordingly. Our proposed methods generally achieve the best accuracy performance across various experiments. Even in cases where they do not outperform all other approaches, they still deliver results that are comparable to the top-performing methods. We observe that `GenMQD` presents a more significant advantage when the size of the initial labeled set is small. This observation demonstrates the potential of our method, as AL scenarios often start with a limited labeled dataset. The ability of our `GenMQD` method to excel in such settings highlights its efficacy in leveraging minimal labeled data to guide the learning process.

Table 1: Accuracy (%) performance of the comparison methods on CIFAR-10 with different sizes of the initial labeled set and validation set. The results of different sizes of the initial labeled set and validation set are reported in the upper and lower parts of the table, respectively. The best accuracy performance of different sizes of validation set are highlighted in boldface.

| Methods | Settings | Number of Queried Examples ($\times 100$) | | | | |
|---|---|---|---|---|---|---|
| | | 1 | 2 | 3 | 4 | 5 |
| Results with Different Sizes of the Initial Labeled Set (in Section 4.3) | | | | | | |
| GenMQD | $n_l = 100$ | 87.95 | 88.27 | 88.32 | 88.17 | 88.27 |
| | $n_l = 500$ | 91.15 | 91.10 | 90.93 | 91.06 | 91.02 |
| | $n_l = 1000$ | 92.61 | 92.43 | 92.31 | 92.33 | 92.34 |
| GenMQD-Fea | $n_l = 100$ | 86.65 | 86.85 | 87.20 | 87.37 | 87.30 |
| | $n_l = 500$ | 90.91 | 90.67 | 90.72 | 90.68 | 90.41 |
| | $n_l = 1000$ | 92.61 | 92.31 | 92.10 | 92.10 | 91.99 |
| RandomText | $n_l = 100$ | 87.76 | 87.92 | 87.65 | 87.77 | 87.73 |
| | $n_l = 500$ | 91.23 | 91.28 | 91.30 | 91.32 | 91.36 |
| | $n_l = 1000$ | 92.63 | 92.52 | 92.47 | 92.42 | 92.47 |
| GAAL | $n_l = 100$ | 85.06 | 84.30 | 84.40 | 84.30 | 83.78 |
| | $n_l = 500$ | 90.57 | 90.63 | 90.45 | 90.20 | 90.34 |
| | $n_l = 1000$ | 92.60 | 92.37 | 92.24 | 92.16 | 92.00 |
| ACGAN | $n_l = 100$ | 83.84 | 83.74 | 83.52 | 83.49 | 83.65 |
| | $n_l = 500$ | 90.48 | 90.10 | 90.26 | 90.44 | 90.49 |
| | $n_l = 1000$ | 92.32 | 92.26 | 91.97 | 91.83 | 91.78 |
| ActiveGAN | $n_l = 100$ | 85.78 | 85.41 | 85.65 | 84.54 | 85.04 |
| | $n_l = 500$ | 90.36 | 90.10 | 90.04 | 90.00 | 90.00 |
| | $n_l = 1000$ | 92.21 | 92.06 | 92.02 | 91.88 | 91.82 |
| Results with Different Sizes of the Validation Set (in Section 4.4) | | | | | | |
| GenMQD | $n_v = 100$ | **87.95** | **88.27** | 88.32 | 88.17 | 88.27 |
| | $n_v = 300$ | 87.62 | 88.17 | 88.25 | 87.92 | 88.13 |
| | $n_v = 500$ | 87.70 | 88.22 | **88.59** | **88.72** | **88.68** |
| GenMQD-Fea | $n_v = 100$ | 86.65 | 86.85 | 87.20 | 87.37 | 87.30 |
| | $n_v = 300$ | 86.78 | 86.87 | 87.08 | 88.14 | 88.04 |
| | $n_v = 500$ | **86.81** | **87.73** | **88.22** | **88.21** | **88.33** |

## 4.4 STUDY ON DIFFERENT SIZES OF THE VALIDATION SET

Our proposed `GenMQD` method utilizes a validation set for query synthesis. Thus, we investigate the impact of its size on model accuracy performance. In addition to the default size of 100 examples, we also evaluate the methods' performance with validation sets of 300 and 500 examples. The best performance is highlighted in boldface. The results are reported in the lower part of Table 1. Extensive experiments on image classification benchmark datasets against query synthesis state-of-the-art methods demonstrate that, on average, `GenMQD` can improve model accuracy by 2.77% on CIFAR-10, and by 2.43% with all datasets, when gathering and labeling 500 examples.

It can be observed that increasing the size of the validation set generally enhances performance, albeit the improvement is marginal. This implies that our `GenMQD` method performs effectively even with a smaller validation set, highlighting its practical applicability. Interestingly, in some cases, a larger validation set slightly reduces accuracy performance, indicating that the choice of validation set size should be made with caution. However, given the small performance differences, we recommend using a moderately small validation set for our `GenMQD` method.

## 4.5 CASE STUDY

Table 2 illustrates some examples of descriptions generated by our proposed `GenMQD` method on CIFAR-10, along with the corresponding images synthesized by the pre-trained Text-to-Image model utilized to simulate the human oracle.

It can be observed that the majority of the generated descriptions are easily comprehensible for human oracles involved in data gathering and labeling, demonstrating the superiority of our method

Table 2: Examples of descriptions generated by our proposed `GenMQD` method and the corresponding images on CIFAR-10. The class associated with each query is highlighted in boldface.

| Generated Text Descriptions | Collected Images |
| --- | --- |
| An image associated with **airplane**, which illustrates a commercial airplane sitting on the runway of an airport. |  |
| An image associated with **automobile**, which illustrates that a car is in a busy street with several cars. |  |
| An image associated with **bird**, which illustrates that a bird is perched on a small branch. |  |
| An image associated with **cat**, which illustrates that a cat is sitting on the ground looking at another cat. |  |
| An image associated with **deer**, which illustrates that a giraffe is standing in the wild near some grass. |  |
| An image associated with **dog**, which illustrates that a dog is on the ground with another dog looking at it. |  |
| An image associated with **frog**, which illustrates that a person is on some kind of food. |  |
| An image associated with **horse**, which illustrates that a horse is standing in the middle of some horses. |  |
| An image associated with **ship**, which illustrates a number of boats docked in the water by a boat. |  |
| An image associated with **truck**. Specifically, it illustrates that a truck is driving on a large road near other vehicles. |  |

in practice. Notably, many prompts describe multiple objects for querying, which reflects a high level of informativeness. Given that most training images in CIFAR-10 contain only a single object, the model lacks knowledge about images featuring multiple objects. We also acknowledge that `GenMQD` encounters limitations in certain cases, such as with the classes involving deers and frogs. These instances reveal gaps in the model's knowledge. For example, the prompt associated with deer shows that the model confuses deer with giraffes. This confusion still offers valuable guidance for the human oracle in gathering data for which the model is uncertain.

These results demonstrate that the proposed `GenMQD` method can mine the desired data from the input space, thereby circumventing the need for an unlabeled data pool. The generated textual descriptions convey semantic information and are more comprehensible for human oracles, thereby facilitating their understanding of and responses to the queries.

## 4.6 USER STUDY

To investigate the interpretability of the textual descriptions generated by `GenMQD` by human oracles, we conduct a user study involving 340 participants who are eligible in annotating image classification data. They are asked to make a choice between *textual description queries* and *generated images* to determine which can be answered with a higher level of confidence. To implement this, the online survey platform randomly samples from the set of texts generated by `GenMQD` and images generated by ACGAN for each given scenario. The positioning of these two choices for each question is randomized for each instance of the survey. The participants are then asked to choose the query type which is easier for providing accurate answers. The survey is conducted using a large question bank. In each survey, 10 questions are randomly selected for each participant. Importantly, the selection process is designed to ensure that each question has a similar or nearly equivalent number of participants answering it. An example survey question is shown in Fig. 3 and the statistics of the results are summarized in Fig. 4.

Table 3: Random sampling statistics (10 times) for survey responses

|  | 1 | 2 | 3 | 4 | 5 | 6 | 7 | 8 | 9 | 10 |
|---|---|---|---|---|---|---|---|---|---|---|
| Sample mean | 25.0 | 26.8 | 26.4 | 27.6 | 24.8 | 25.0 | 27.0 | 27.2 | 27.2 | 26.6 |
| Sample variance | 22.5 | 3.2 | 2.8 | 2.8 | 21.2 | 22.5 | 2.5 | 3.2 | 2.7 | 3.3 |
| t-statistic | 3.78 | 12.25 | 12.56 | 14.16 | 3.79 | 3.77 | 14.14 | 12.75 | 13.88 | 11.82 |
| p value ($\times 1e-4$) | 195.84 | 2.55 | 2.31 | 1.44 | 193.02 | 195.84 | 1.45 | 2.18 | 1.56 | 2.94 |

Which of the following queries can you answer with a higher level of confidence?

Please find and upload some images about the scenario of "A truck is driving on a large road near other vehicles ."

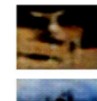
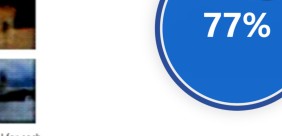
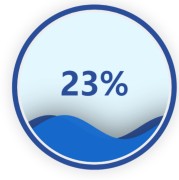

○ Based on this text, fufill the objective

○ Please select the most accurate label for each image: 'plane', 'car', 'bird', 'cat', 'deer', 'dog', 'frog', 'horse', 'ship', 'truck'

**77%**

**23%**

(a) Text Query    (b) Image Query

Figure 3: An example survey question.

Figure 4: User preference.

Among the survey responses received, 77% users prefer textual queries over generated images. Table 3 reports statistical significance test results for the our user study. The sample mean values for correct answers across the 10 random samples range from 24.8 to 27.6, significantly exceeding the hypothesized benchmark of 17, corresponding to a 50% correctness rate. This indicates that participants performed consistently better than random guessing. Sample variances range from 2.5 to 22.5, with smaller variances suggesting more consistent performance across participants, while larger variances reflect greater fluctuation in the number of correct answers. The t-statistics, ranging from 3.77 to 14.16, underscore the significant difference between the sample means and the hypothesized value of 17, confirming that participants' correctness rates are notably higher. Furthermore, all p-values fall below the 0.05 threshold, indicating statistical significance for each sample. This demonstrates that participants' performance is not due to chance but is significantly better than the expected 50% correctness rate. In conclusion, this analysis shows that, even with questions randomly selected from a large database, participants exhibit a strong understanding of the material, with correctness rates consistently higher than random guessing. Both the mean values and statistical tests confirm the reliability of this conclusion.

This finding underscores the significant advantage of our proposed `GenMQD` method compared to the prevailing querying with generated images AL methods. As demonstrated in the image query example, some generated instances are blurry and difficult for human evaluators to recognize. This limitation arises from the query synthesis methods, because the initial labeled set in AL is usually small, and there is no available pool of unlabeled data for synthesis methods. Thus, GAN-based models struggle to be effectively trained. Under this challenging scenario, our proposed `GenMQD` method mitigates the risk of querying humans with unrecognizable data by describing the desired data using natural language. This approach can enhance the human oracle's understanding of the active learner's needs, thereby facilitating more effective data gathering and labeling.

## 5 CONCLUSIONS AND FUTURE WORK

In this paper, we propose a novel active querying method, `GenMQD`, which aims to generate the textual descriptions of desired data for AL data gathering and labeling. Unlike traditional methods, it does not rely on an unlabeled data pool for instance selection, making it more applicable in practice. `GenMQD` first synthesizes the desired data by optimizing model performance on a validation set. We show that this problem can be efficiently solved using influence functions. Subsequently, it generates textual descriptions of the synthesized data with a pre-trained multi-modal alignment model. Extensive experiments demonstrate that the proposed `GenMQD` method effectively generates informative data and semantically aligned textual descriptions for querying. Furthermore, a large-scale user study reveals that this active query type is more comprehensible to human oracles. In future, we plan to investigate alternative text decoding techniques to produce more detailed descriptions.

ETHICS STATEMENT

In this section, we acknowledge the potential ethical considerations associated with the development and application of `GenMQD`. Our approach involves synthesizing data and generating textual descriptions, which could introduce biases if the multi-modal alignment model or the influence functions used carry any underlying biases. Additionally, the user study conducted to evaluate the comprehensibility of the generated queries relied on human subjects, and we took care to ensure that participants gave informed consent and their data was handled with strict confidentiality. As we move forward, we recognize the importance of continuously evaluating and mitigating any risks related to model bias, transparency, and data privacy. Responsible innovation is at the core of our research, and we are committed to creating tools that empower users without compromising ethical standards.

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

# A  APPENDIX

In the appendix, we provide a more comprehensive and detailed explanation of the proposed methodology, along with additional experimental results. This section is organized to offer a deeper understanding of each aspect of our study, making the information more accessible to the reader. The content is structured as follows:

- First, we present a detailed description of the `GenMQD`-Fea method, outlining its specific components, with a focus on the provided pseudo-codes that illustrate the algorithmic workflow. Additionally, we include an analysis of the method, emphasizing the rationale behind its design choices.

- Next, we delve into the experimental results, providing additional insights beyond what was initially discussed in the main text. This includes the learning curve plots that show the model's performance when trained and validated with varying sizes.

- Finally, we provide further elaboration on the user study, including the design methodology of the survey and concrete examples of the questions posed to participants.

## A.1  PSEUDO CODES OF `GenMQD`-FEA ALGORITHM

In our experiments, we introduce and evaluate a variant of our proposed method, termed `GenMQD`-Fea, as part of a comprehensive performance comparison. The specific details of this variant are illustrated in the pseudo-code provided in Algorithm 2. The key difference between the standard `GenMQD` method and its `GenMQD`-Fea variant lies in the fact that the latter eliminates the text generation phase. Instead of synthesizing text-based data, the `GenMQD`-Fea variant directly leverages a synthesized data point, denoted as $z^*$, for model updates.

The generation of the synthesized data point $\boldsymbol{x}^*$ is intricately dependent on the pre-selected class label $y^*$. By optimizing $\boldsymbol{x}^*$, we implicitly enhance the model's ability to generalize, as this optimization aligns the model more closely with the underlying patterns in the validation set. Consequently, the pseudo-label $y^*$ assigned to $\boldsymbol{x}^*$ tends to be more accurate and trustworthy when compared to labels produced by the compared data synthesis techniques. This improved reliability of pseudo-labels contributes to the overall performance of the model. Furthermore, the results of the performance comparison in our experiments provide strong empirical evidence that the `GenMQD`-Fea method achieves superior outcomes, demonstrating both its effectiveness and its potential for broader applications in similar tasks.

---

**Algorithm 2** The Proposed `GenMQD`-Fea Algorithm

---

**Input:** Labeled set $L$, validation set $V$, pre-trained CLIP model, classification model parameter $\boldsymbol{\theta}$, optimization epochs $T$, learning rate $\alpha$, query batch size $n_q$.
**Output:** Active collected dataset $Q$.

1: $Q \leftarrow \varnothing$
2: $\{\boldsymbol{x}_i \in L \cup V\} \leftarrow$ extract features for all images using a pre-trained CLIP image encoder.
3: **for** $y^* = \{1, \ldots, |\mathcal{Y}|\}$ **do**
4:      $\boldsymbol{x}^* \leftarrow \text{Mean}(\{\boldsymbol{x}_i \in L \mid y_i = y^*\})$ ▷ *Initialize a feature vector using the class center of $y^*$.*
5:      **for** $j = 1, \ldots, T$ **do**
6:          $\hat{\boldsymbol{\theta}} \leftarrow$ get the empirical risk minimizer on $L \cup \{\boldsymbol{x}^*, y^*\}$.
7:          $\delta \leftarrow \frac{1}{n_v} \sum_{z_i \in V} \left[ -\nabla_{\boldsymbol{\theta}} \ell \left( z_i, \hat{\boldsymbol{\theta}} \right)^\top H_{\hat{\boldsymbol{\theta}}}^{-1} \nabla_{\boldsymbol{x}^*} \nabla_{\boldsymbol{\theta}} \ell \left( z^*, \hat{\boldsymbol{\theta}} \right) \right]$       ▷ *By Eq. (5)*
8:          $\boldsymbol{x}^* \leftarrow \boldsymbol{x}^* + \alpha \delta$
9:      **end for**
10:      $z^* \leftarrow \{\boldsymbol{x}^*, y^*\}$
11:      Add $z^*$ to $Q$
12: **end for**
13: **return** $Q$

---

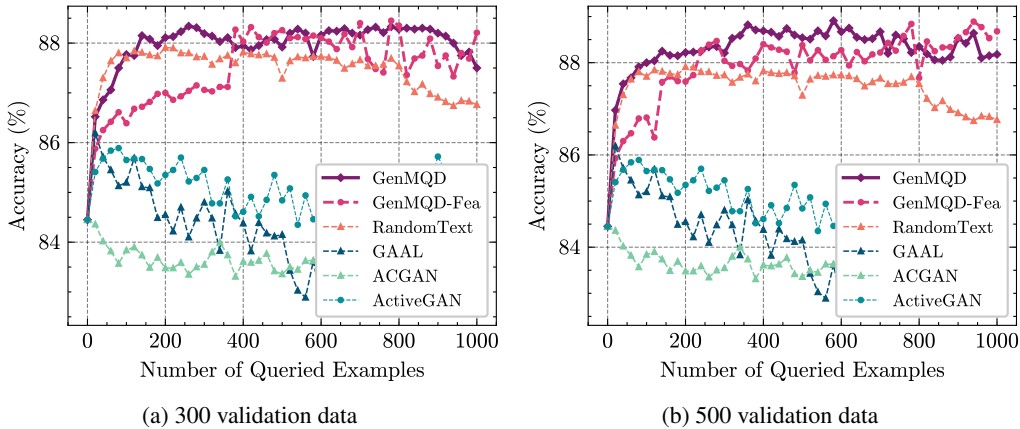

(a) 300 validation data                    (b) 500 validation data

Figure 5: Performance comparisons between baseline methods and our proposed methods with different sizes of validation set.

## A.2  Performance Comparisons with Different Sizes of Validation Set

In Section 4, we present an evaluation of the performance of our proposed methods when tested with varying sizes of the validation set in tabular form. To provide a more in-depth and dynamic analysis of the model behavior, we also include the complete learning curves, which visually illustrate the progression of performance over time. These results are illustrated in Fig. 5.

As can be observed from the figure, both the `GenMQD` and `GenMQD`-Fea methods exhibit improvements in performance as the size of the validation set increases. The additional information provided by the larger validation set appears to enable both methods to generalize better and refine their predictions more effectively. Specifically, in comparison to the performance observed with only 100 validation samples, which is demonstrated in Fig. 2, the `GenMQD`-Fea method shows a more significant and consistent performance advantage over the RandomText baseline when additional validation data becomes available.

Moreover, despite the differences in validation set sizes, the overall performance gap between `GenMQD` and `GenMQD`-Fea remains consistently narrow. This indicates that even when operating with a relatively small validation set, both methods maintain a high level of performance and efficiency. The small gap further suggests that `GenMQD`-Fea, while slightly more advantageous with larger validation sets, does not exhibit a drastic drop-off in performance when data is more limited. This finding highlights the robustness and flexibility of our methods, suggesting that they can be applied effectively in practical scenarios where access to large validation sets may be constrained.

## A.3  Details of User Study

In this section, we provide further details and examples from the user study. More examples of the survey is illustrated in Fig. 6.

The bank of text query options consists of descriptions generated by `GenMQD` during AL iterations 1 through 100. In each iteration, `GenMQD` produces a description corresponding to the desired data for each class. As a result, the bank contains an equal number of descriptions for each class. For image generation, we use a trained ACGAN model to generate an equivalent quantity of data for each class. During the user study, we randomly select samples from the bank and query the preference from the users.

Which of the following queries can you answer with a higher level of confidence?

# Please find and upload some images about the scenario of "A cat is leaning on the ground looking at a cat"

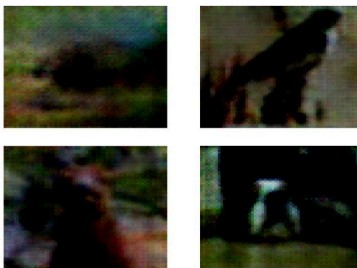

○ Based on this text, fufill the objective

○ Please select the most accurate label for each image: **'plane', 'car', 'bird', 'cat', 'deer', 'dog', 'frog', 'horse', 'ship', 'truck'**

Which of the following queries can you answer with a higher level of confidence?

# Please find and upload some images about the scenario of "A horse is standing in the middle of some horses"

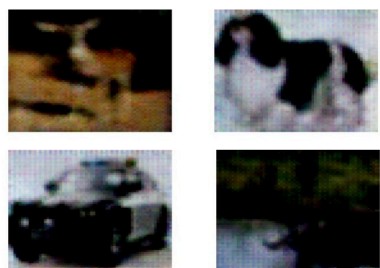

○ Based on this text, fufill the objective

○ Please select the most accurate label for each image: **'plane', 'car', 'bird', 'cat', 'deer', 'dog', 'frog', 'horse', 'ship', 'truck'**

Which of the following queries can you answer with a higher level of confidence?

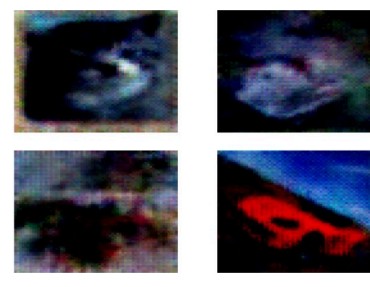

# Please find and upload some images about the scenario of " A dog is on the ground with a dog nearby"

○ Please select the most accurate label for each image: **'plane', 'car', 'bird', 'cat', 'deer', 'dog', 'frog', 'horse', 'ship', 'truck'**

○ Based on this text, fufill the objective

Figure 6: Examples of the survey in user study.