# OpenReview forum: "Semantic-aligned Query Synthesis for Active Learning"
_ICLR.cc/2025/Conference — Submitted to ICLR 2025_

### Official Review · Reviewer_UqAq · 2024-10-26

**Soundness:** 3
**Presentation:** 3
**Contribution:** 3
**Rating:** 5
**Confidence:** 5

**Summary:**

This paper proposes the Generative Membership Query Descriptor (GenMQD), a novel approach for generating textual descriptions of target data. The proposed method leverages the pre-trained CLIP model to capture optimal textual descriptions, which are then used by Stable Diffusion to generate corresponding data. Extensive experiments validate the effectiveness of the proposed GenMQD.

**Strengths:**

1. The paper considers a practical scenario where the pool of unlabeled data may be limited.

2. The paper is written in a clear and accessible manner.

3. The paper optimizes $x^*$ effectively through the influence function.

4. The large-scale user study is impressive, clearly demonstrating a preference for the proposed text query.

**Weaknesses:**

1. We cannot guarantee that $v_x^* (= t_x^*)$ is the same with ImgEncode(StableDiffusion(DeCap($t_x^*$))). It is necessary to evaluate the cosine similarity or distance between these embeddings.

2. It is difficult to confirm whether the experimental results are caused by the most important components $x^*$ in the proposed method. GenMQD outperforms RandomText, which uses only class names, but it should also be compared to using sentences generated by LLMs that simply describe the class.

3. The experiments were conducted solely on datasets with a limited number of classes.

**Questions:**

1. Figure 1 seems to be inconsistent with the experiments and content of the paper. Requesting some examples from a person implies a manual search through the unlabeled image pool, yet the paper utilizes Stable Diffusion, doesn't it?

2. It seems unlikely that actual humans were involved during the experiments, so I'm curious about how labels were assigned to the synthetic images. I wonder the paper is about active learning with humans or if it leans more toward an automated labeling process.

---

### Official Review · Reviewer_k4L8 · 2024-11-02

**Soundness:** 3
**Presentation:** 2
**Contribution:** 3
**Rating:** 5
**Confidence:** 5

**Summary:**

Active learning aims to maximize the model performance with the constrained labeling budget. This paper introduces the Generative Membership Query Descriptor (GenMQD) method for active learning, which generates textual descriptions instead of directly sample selection. By leveraging multi-modal models like CLIP to convert these descriptions into natural language, GenMQD enhances model accuracy by 2.43% and is preferred by human annotators over traditional image-based queries.

**Strengths:**

**[Novelty]** In the previous works of active learning, most of works were focused on how to select the informative dataset to enhance the model significantly. However, this paper focused on how to generate the informative text to gather the corresponding images. I think this concept makes sense, and can be a novel approach.

**[Thorough Analysis]** My major concern is that *Is the description well generated, and is there any corresponding images?*. As shown in Tab. 2, the generated texts seem to be meaningful, and these are supporting evidences for improving the model performance.

**Weaknesses:**

**[Baselines]** I think this method should be compared with not only query / data synthesizing methods but also sample selection methods that is traditional AL techniques. This is because sample selection is currently the major stream in active learning research domain.

**[Prove the effectiveness]** Tab. 1 shows that the proposed method has lower performance than the other baselines, which seem to be a different setting. However, in the same setting, there is no other methods to compare with proposed method. I think several baselines should be added in the same setting.

**[Unfamiliar datasets for CLIP]** The authors adopted the pre-trained CLIP for extracting image features and generating the text from the features. However, there are several datasets that are not familiar with CLIP such as Flowers102, EuroSAT, etc. I'm wondering that *This method is applicable for all the datasets? Or is it only applicable for specific datasets that are familiar with CLIP?*. In former one, I think the PEFT method (e.g, LoRA) can be applied into CLIP.

**[Related Works]** It seems to be that Related Work section needs to be enhanced. I suggest the recent AL papers as below:

[1] Active Prompt Learning in Vision Language Models, CVPR 2024

[2] Active Generalized Category Discovery, CVPR 2024

[3] Entropic Open-Set Active Learning, AAAI 2024

Overall, it seems to be good approach, but there are several concerns in this work. I am going to raise my score if the concerns are solved.

**Questions:**

I already stated in the weakness section.

---

### Official Review · Reviewer_wpBM · 2024-11-03

**Soundness:** 2
**Presentation:** 2
**Contribution:** 2
**Rating:** 3
**Confidence:** 4

**Summary:**

This paper investigates active learning in the context of reducing data labeling costs through synthetic data generation. The authors propose a novel approach that first retrieves text descriptions from an embedding space and then leverages these descriptions to generate corresponding training images. The effectiveness of this methodology is validated through experimental evaluations.

**Strengths:**

1. The paper addresses a significant challenge in active learning by exploring synthetic data generation as an alternative to costly human labeling.
2. The proposed approach of leveraging generative models for data synthesis presents an innovative solution to the data labeling bottleneck.

**Weaknesses:**

1. The literature review lacks comprehensive coverage of recent developments in synthetic data generation for training, particularly from the past three years.
2. The comparative analysis relies on outdated baselines, with ActiveGAN (published 5 years ago) being the most recent comparison. More recent approaches are suggested to add.
3. The improvement is mild. The proposed method uses the knowledge from the CLIP model, which is pretrained on a large-scale data. However, the zero-shot of the CLIP model on CIFAR-10 can achieve 91.3 Acc, whereas the proposed method has similar performance but using additional resources.
4. The computational complexity analysis is insufficient, particularly regarding data generation time. Critical computational bottlenecks include: 1) Hessian matrix calculation; 2) Image generation via stable diffusion. These components likely incur significant computational overhead as sample size, image resolution, and hyperparameter optimization epochs T increase.

**Questions:**

Please see the weakness above.

---

### Official Review · Reviewer_gFq8 · 2024-11-03

**Soundness:** 3
**Presentation:** 4
**Contribution:** 3
**Rating:** 5
**Confidence:** 4

**Summary:**

This paper leveraged the influence function to generate certain synthetics data which has semantic meaning, and using the generated data to query the most relevant sample in the validation set to improve the test performance.

**Strengths:**

1. This paper is well written and well organized.
2. The idea of generating the most informative data from the influence of perturbation is interesting and novel.

**Weaknesses:**

1. I have concerns on the influence of perturbation, as we can find from the Eq. 5 to Eq. 3, it seems that you are looking for the perturbation of $x^*$ which can most attach to the validation set, in other word, you are looking for the data which should mostly fit into the training and validation distribution. However, since both training and validation set you claimed is very small, therefore, I have little concern on whether this would lead the training to direct involve with the validation set (loss computation of Eq. 5) instead of active querying on the validation set as previous works.
2. It seems the generation process of the $\delta$ would requires a lot iteration steps, I have concerns on the computation overhead.
3. Some benchmark missed, e.g. CIFAR-100
4. The fundamental implementation of this work replies heavily on the pretrained networks, which introduce the prior knowledge of the data  and is one thing I concern, is other models your used are also pretrained, e.g. GAN.
5. Some notations and demonstrations are ambiguous, e.g. $p(\cdot)$ process in Algorithm 1, line 11, is the $\bold{t}$ is the matched text results for the $\bold{x}^*$? So is $\bold{t}$ are generated from the images (the previous text embeddings from train)or just calculated from Eq.6, if so, why the $\bold{t}$ is not equal to the $\bold{v}_{\bold{x}^*}$? Why it is necessary to mention Eq.6?

**Questions:**

See weakness

---

### Official Review · Reviewer_kBte · 2024-11-03

**Soundness:** 2
**Presentation:** 2
**Contribution:** 2
**Rating:** 3
**Confidence:** 4

**Summary:**

Unlike previous approaches in active learning that focus on selecting informative samples for labeling, this paper addresses the data collection phase. To achieve this, datasets can be generated using generative models. However, this approach has significant drawbacks as it can produce uninformative or even irrelevant images. To address this issue, the authors propose the Generative Membership Query Descriptor (GenMQD). The main idea of this paper is to generate descriptions of the desired samples rather than the samples themselves. The authors demonstrate performance improvements across various evaluation settings.

**Strengths:**

1. Tackles an interesting research problem

2. Shows performance improvements

3. Conduct a user study, which is a costly experimental approach

**Weaknesses:**

1. Although the authors acknowledge that the performance difference between GenMQD and GenMQD-Fea across datasets is an open question, this issue must be addressed.

2. Given the marginal performance differences across entities, a random-seed analysis should be conducted. Without this, it is challenging to interpret the impact of "the number of query examples" and $n_l$ or $n_v$ in Table 1.

**Questions:**

Please refer to the Weakness part.

---

### Meta-Review · Area_Chair_UoC5 · 2024-12-24

**Metareview:**

**Summary:** This paper addresses active learning in scenarios with limited unlabeled data by proposing the Generative Membership Query Descriptor (GenMQD) algorithm. Instead of synthesizing data samples directly, GenMQD generates textual descriptions of the desired data, which are then transformed into natural language queries using pre-trained multi-modal models like CLIP. The motivation is to mitigate the risk of generating uninformative or irrelevant samples, a common issue with existing methods. Empirical evaluations on the CIFAR-10 and iNaturalist data sets demonstrate the improvements of GenMQD over several baseline algorithms, and a user study shows that human annotators prefer the generated queries over image-based ones.

**Decision:** Despite tackling an interesting and practical problem, the paper has several shortcomings that lead to the decision to reject.  Specifically, reviewers are concerned that the experimental validation is insufficient, relying on limited datasets, and outdated or weak baselines. The method’s reliance on computationally expensive processes, such as Hessian calculations and stable diffusion, makes it impractical for real-world applications (gFq8, wpBM). Additionally, the heavy dependency on pre-trained models like CLIP limits the generalizability of the approach to datasets unfamiliar to these models (gFq8, wpBM, k4L8). Methodological novelty is limited, as the work primarily integrates existing approaches without substantial innovation (wpBM). Finally, ambiguities in the algorithm and an incomplete related work section further detract from the overall clarity and impact of the paper. These limitations outweigh the contributions, leading to the decision to reject. During the reviewer-AC discussion period, the reviewers unanimously agreed with this decision.

**Additional Comments On Reviewer Discussion:**

The authors did not respond to the reviewers' comments during the rebuttal period.

---

### Decision · Program_Chairs · 2025-01-22

Reject